# Cell culture-based karyotyping of orectolobiform sharks for chromosome-scale genome analysis

Yoshinobu Uno [1,6 ✉], Ryo Nozu [2,3], Itsuki Kiyatake[4], Nobuyuki Higashiguchi[5], Shuji Sodeyama[4], Kiyomi Murakumo[3], Keiichi Sato[2,3] & Shigehiro Kuraku[1]

Karyotyping, traditionally performed using cytogenetic techniques, is indispensable for validating genome assemblies whose sequence lengths can be scaled up to chromosome sizes using modern methods. Karyotype reports of chondrichthyans are scarce because of the difficulty in cell culture. Here, we focused on carpet shark species and the culture conditions for fibroblasts and lymphocytes. The utility of the cultured cells enabled the high-fidelity characterization of their karyotypes, namely $2n = 102$ for the whale shark (*Rhincodon typus*) and zebra shark (*Stegostoma fasciatum*), and $2n = 106$ for the brownbanded bamboo shark (*Chiloscyllium punctatum*) and whitespotted bamboo shark (*C. plagiosum*). We identified heteromorphic XX/XY sex chromosomes for the two latter species and demonstrated the first-ever fluorescence in situ hybridization of shark chromosomes prepared from cultured cells. Our protocols are applicable to diverse chondrichthyan species and will deepen the understanding of early vertebrate evolution at the molecular level.

[1] Laboratory for Phyloinformatics, RIKEN Center for Biosystems Dynamics Research (BDR), Kobe, Japan. [2] Okinawa Churashima Research Center, Okinawa Churashima Foundation, Okinawa, Japan. [3] Okinawa Churaumi Aquarium, Okinawa, Japan. [4] Osaka Aquarium Kaiyukan, Osaka, Japan. [5] Aquament Co., Ltd, Kobe, Japan. [6] Present address: Department of Life Sciences, Graduate School of Arts and Sciences, The University of Tokyo, Tokyo, Japan. ✉ email: unoy@g. ecc.u-tokyo.ac.jp

Recent improvements in long-read sequencing technology and Hi-C, a genome-wide chromosome conformation capture technology, have enabled the assembly of many complex eukaryotic genomes for reconstructing chromosome-scale sequences[1-3]. Previously, this was achieved only by genetic linkage mapping, performed with a number of offspring[4]. In controlling resultant chromosome-scale sequences, karyotyping serves the final goal by providing information about the number of chromosomes. However, among vertebrates, the karyotype of species within Chondrichthyes (cartilaginous fishes) is the least investigated because of the lack of a reliable protocol for chromosome preparation using cultured cells[5,6]. This limitation has been observed for the elephant fish *Callorhinchus milii* (also called elephant shark), the first chondrichthyan species with a sequenced genome in the absence of karyotype information[7]. This species is used heavily for in silico sequence analysis, but not in other life sciences, especially those conducted in the laboratory. Chondrichthyan cell culture does not only provide functional validation but also produces high-quality chromosome spreads for karyotyping and molecular cytogenetic analyses. Crucially, because of body fluid osmolality peculiar to chondrichthyans, the technical difficulties inherent in cell culture have prevented karyotype investigation of chondrichthyan species[8].

The class Chondrichthyes occupies a unique phylogenetic position as a sister group to all other jawed vertebrates and comprises two extant subclasses, Elasmobranchii (sharks, rays and skates) and Holocephali (chimeras)[9]. The former contains 13 orders that comprise over 1200 species, and the latter contains one order that includes about 60 species[10]. Orectolobiformes is the third most species-rich order of sharks and comprises 45 species that inhabit mainly temperate or tropical waters in the Pacific Ocean. Within this order, the whale shark *Rhincodon typus* (Fig. 1) is unique as a pelagic species with the largest body size as a 'fish' (reviewed in ref. [11]). Several chondrichthyans, including this large-bodied species, have been subjected to whole-genome sequencing[7,12-14]. However, the sequencing output cannot be validated without the goal of sequencing, namely karyotype information that provides the inherent number and size of chromosomes.

Among Chondrichthyes, karyotypes have been reported for 83 of the ~1300 known species, namely 81 elasmobranch and two holocephalan species (as of May 2020), most of which have 50–86 chromosomes[5,6,15,16] (Supplementary Data 1). However, to our knowledge, no reliable karyotype reports are available for Orectolobiformes. In most past cytogenetic studies of chondrichthyans, chromosome preparations were prepared using in vivo treatment protocols involving the collection of mitotic cells directly from animal tissues in which mitotic inhibitors were injected before sacrifice[17,18]. The abundance of chromosomes has been a hurdle in chondrichthyan cytogenetics, but the most crucial obstacle lies in the supply of cultured cells to obtain high-quality chromosome spreads.

Cell culture is an important tool for studies using traditional laboratory animals such as the mouse and chicken. When applied to chromosome studies, cell culture facilitates the preparation of high-quality chromosome spreads with high metaphase frequency, as exemplified by the karyotyping of diverse species, including one with more than 150 chromosomes[19], and by high-throughput chromosome mapping using fluorescence in situ hybridization (FISH)[20]. However, it has remained to be stably applied to chondrichthyans whose blood osmolality is approximately three times higher than that of mammalians and teleost fishes[8]. For example, the culture medium used for marine teleost fishes cannot be readily applied to chondrichthyans. The medium formulations must be optimized because of the high osmolality required for cell culture from any tissue (except for early embryos) of chondrichthyans. As a result, the optimal conditions

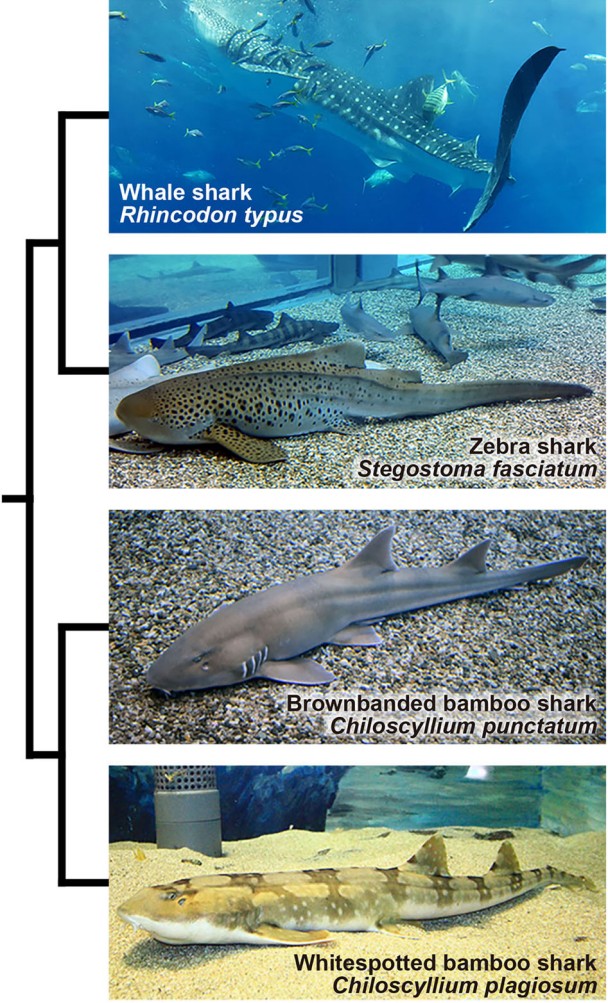

**Fig. 1 Four orectolobiform shark species analysed in this study.** The sizes of the animals are not to scale. The phylogenetic relationship between these species is based on the existing literature[9]. The branch lengths do not reflect phylogenetic distance.

for the culture of cells for karyotyping and FISH mapping have not been established for chondrichthyans.

Fibroblast and lymphocyte cultures of chondrichthyan tissues have been shown to require the supplementation of urea and NaCl to the culture medium[21-28] (Supplementary Data 2). Multipassage fibroblast culture for cartilaginous fish has been documented only for the spiny dogfish shark *Squalus acanthias* (later designated *Squalus suckleyi*), in which the addition of cell growth factors and shark yolk extract allowed a continuously proliferating cell line[29]. These growth factors have also been used for fibroblast culture of teleost fishes[30,31]. For lymphocyte culture, mitogens are the most important factors affecting the mitotic index and trigger a polyclonal proliferation of lymphocytes through blastoid transformation. Lymphocytes of the nurse shark *Ginglymostoma cirratum* respond to stimulation by concanavalin A (Con A), lipopolysaccharide (LPS) and phytohaemagglutinin (PHA)[32,33]. In the previously reported chondrichthyan lymphocyte culture, PHA and phorbol 12-myristate 13-acetate (PMA) were used as mitogens to increase the endogenous mitotic activity[24,26] (Supplementary Data 2). Among these four reagents, Con A and PHA are used as T cell mitogens, and LPS and PMA are used as B cell mitogens in mammalian cell culture. To date, there are no reports of the use of Con A and LPS as mitogens for cell culture of chondrichthyans.

In this study, we have derived protocols for cell culture of fibroblasts and lymphocytes for four orectolobiform shark species, the whale shark, zebra shark *Stegostoma fasciatum* (also known as *S. tigrinum*[34]), brownbanded bamboo shark *Chiloscyllium punctatum*, and whitespotted bamboo shark *C. plagiosum* (Fig. 1). Using the cultured cells, we have revealed the karyotypes and performed FISH mapping for these species. Moreover, we have demonstrated the potential utility of our method for modern genomic studies.

## Results

**Species identification of the bamboo sharks**. To distinguish the two *Chiloscyllium* species from their closely related species (Fig. 1), partial DNA fragments of the mitochondrial cytochrome oxidase subunit 1 (*COX1*) gene were cloned, and their nucleotide sequences were determined. The obtained nucleotide sequences of the 325-bp-long DNA fragment of the brownbanded bamboo shark (NCBI GenBank Accession ID: LC537758) and whitespotted bamboo shark (LC537759) showed 99.4–100% identity (difference of no more than two nucleotides) to those of the respective species (JN313263 and MG574425) deposited in the NCBI Nucleotide database. These DNA sequences in NCBI exhibited a difference of 24 nucleotides in the selected *COX1* region between the two species, which provided firm evidence of our species determination.

**Cell culture from shark tissues**. For fibroblast culture, we derived culture medium supplemented with urea, NaCl and three kinds of cell growth factors (insulin-transferrin-selenium [ITS-G], epidermal growth factor and fibroblast growth factor) at the concentrations used in previous studies[23,25,29,31] (Supplementary Fig. 1 and Supplementary Data 2). For lymphocyte culture, we adopted culture medium supplemented with the combination of four mitogens, Con A, LPS, PHA and PMA, at the concentrations used in previous studies[24,35] (Supplementary Fig. 1 and Supplementary Data 2).

For the multipassage culture of fibroblasts, we performed cell culture from whole embryos and juvenile tissues of the two bamboo shark species (Table 1). Outgrowth of fibroblast-like and epithelial-like cells was observed around the tissue fragments of whole embryos and juvenile tissues within a week (Fig. 2a). The cultured cells achieved cellular confluence in primary culture within a month, after which they were dissociated in shark phosphate-buffered saline (SPBS)[26] supplemented with dispase every 2–10 days before subculturing. Fibroblast-like cells prevailed after a few passages (Fig. 2b). Proliferation of fibroblasts was observed in the samples of all individuals analysed.

For lymphocyte culture, we used whole blood from juveniles of the whale shark and adults of the zebra shark and the brownbanded bamboo shark, and spleen from a male juvenile of the whitespotted bamboo shark (Table 1). Cell proliferation was observed from two of four, two of six and all seven individuals of the whale shark, the zebra shark and the two bamboo shark species, respectively (Fig. 2c).

**Shark karyotyping using cultured cells**. To avoid chromosomal aberrations caused by cryopreservation and repeated cell culture, we used fibroblasts from no later than the seventh passages and lymphocytes from primary culture. Chromosome metaphase spreads were prepared from cultured cells from all individuals for which cell proliferation was detected (Fig. 2d and Table 1). Karyotypes were examined for at least 25 metaphases from both sexes of each of the four orectolobiform shark species (Supplementary Data 3).

For the whale shark, blood samples from two male and two female individuals were used for lymphocyte culture (Table 1), of which mitotic chromosomes and cell proliferation were obtained for one male and one female. The diploid chromosome number of this species was found to be 102, which included 16 metacentric or submetacentric, 4 subtelocentric and 82 acrocentric chromosomes (Fig. 3a, Supplementary Fig. 2a and Supplementary Fig. 3a, b). We succeeded in obtaining chromosome spreads from one of three males and one of three females of the zebra shark (Table 1) and found the diploid number of 102, which included 20 metacentric or submetacentric, four subtelocentric, and 78 acrocentric chromosomes (Fig. 3b, Supplementary Fig. 2b and Supplementary Fig. 3c, d). No evident heteromorphic sex chromosomes were detected in either the whale shark or zebra shark.

We collected chromosome spreads from nine males and five females of the brownbanded bamboo shark, and from four males and two females of the whitespotted bamboo shark (Table 1). The brownbanded bamboo shark was found to contain 106 chromosomes, which included 52 metacentric or submetacentric and 54 subtelocentric chromosomes (Fig. 3c, Supplementary Fig. 2c and Supplementary Fig. 3e, f). Of those, one smallest-sized chromosome was observed only in males and was presumed to be a Y chromosome. This suggests that the brownbanded bamboo shark has heteromorphic XX/XY sex chromosomes. However, no X chromosome was unambiguously detected, probably because multiple chromosomes including a putative X chromosome have similar sizes. Large secondary constrictions were observed on the subtelomeric regions of one pair of middle-sized submetacentric chromosomes (Fig. 3c and Supplementary Fig. 2c). The diploid chromosome number of the whitespotted bamboo shark was found to be 106, which included 50 metacentric or submetacentric, 28 subtelocentric and 28 acrocentric chromosomes (Fig. 3d, Supplementary Fig. 2d and Supplementary Fig. 3g, h). Secondary constrictions were observed in the subtelomeric regions of three and four middle-sized submetacentric chromosomes in all males and all females, respectively (Fig. 3d and Supplementary Fig. 2d). This result indicates that the whitespotted bamboo shark karyotypes contain middle-sized submetacentric X chromosomes with secondary constrictions and small-sized putative Y chromosomes without secondary constrictions.

**Localization of 18S–28S rDNA and telomeres using FISH**. To provide a technical demonstration of chromosome mapping by FISH using the successfully cultured cells, we examined the chromosomal distribution of the 18S–28S rRNA genes. We analysed five males and three females of the brownbanded bamboo shark, two males and two females of the whitespotted bamboo shark and one male and one female of the zebra shark (Fig. 4). Intense FISH signals were located in the terminal regions of four middle-sized chromosomes, and weak signals were mapped to one middle-sized chromosome in all males and two females of the brownbanded bamboo shark (Fig. 4a). By contrast, FISH signals were observed on only four chromosomes in the other female (Fig. 4b). In the whitespotted bamboo shark, the 18S–28S rRNA genes were mapped to two middle-sized chromosomes and X chromosomes with secondary constrictions (Fig. 4c, d). However, no FISH signals were detected on putative Y chromosomes in the brownbanded bamboo shark (Fig. 4a) or small-sized chromosomes including putative Y chromosomes in the whitespotted bamboo shark (Fig. 4c). In the zebra shark, the 18S–28S rRNA genes were located on two large-sized submetacentric chromosomes (Fig. 4e).

Fluorescence signals of (TTAGGG)$n$ sequences were observed at the telomeric ends of all chromosomes in the brownbanded bamboo shark, whitespotted bamboo shark, and zebra shark

**Table 1 List of the four shark species and the number of individuals used for cell culture and karyotyping in this study.**

| Species name | No. of individuals used with tissue choice | | | | | |
| --- | --- | --- | --- | --- | --- | --- |
| | Fibroblast culture | | Lymphocyte culture | | Karyotyping | |
| | Male | Female | Male | Female | Male | Female |
| *Rhincodon typus* | | | 2 juveniles (blood) | 2 juveniles (blood) | 1 juvenile | 1 juvenile |
| *Stegostoma fasciatum* | | | 3 adults (blood) | 3 adults (blood) | 1 adult | 1 adult |
| *Chiloscyllium punctatum* | 5 (whole embryo) | 1 (whole embryo) | 3 adults (blood) | 3 adults (blood) | 5 embryos, 3 adults | 1 embryo, 3 adults |
| *Chiloscyllium plagiosum* | 3 (whole embryo) 1 juvenile (kidney, peritoneum) | 2 (whole embryo) | 1 juvenile (spleen) | | 3 embryos, 1 juvenile | 2 embryos |

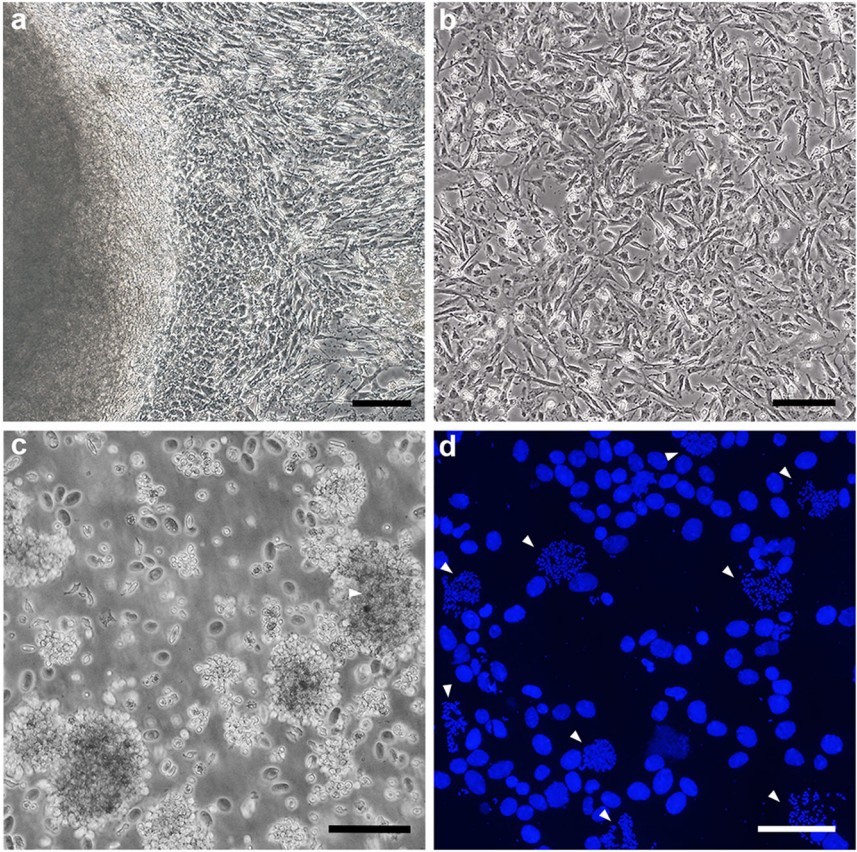

**Fig. 2 Shark cell culture. a** Migration of fibroblast-like and epithelial-like cells in the primary culture from the tissue fragments of a whole embryo of the brownbanded bamboo shark. **b** Fibroblasts from a whole embryo of the brownbanded bamboo shark after seven passages. **c** Aggregated lymphocytes of the whale shark. **d** Typical view of DAPI-stained mitotic cells from fibroblasts of the whitespotted bamboo shark. Arrowheads indicate metaphase chromosome spreads. Scale bars represent 200 μm in **a** and **b**, and 100 μm in **c** and **d**.

(Fig. 4f–h). No interstitial telomeric site was found in these species. We also attempted FISH mapping for the whale shark. However, chromosomal locations of the 18S–28S rRNA genes and telomeric repeats were not confirmed because of an extremely low mitotic index and insufficient quality of chromosome spreads compared with the three other species studied.

**Comparative genomic hybridization (CGH) patterns between male and female chromosomes**. To identify sex-specific chromosomal regions, we performed CGH using metaphase chromosomes from two individuals per sex in the brownbanded bamboo shark and whitespotted bamboo shark in which we observed

heteromorphic sex chromosomes. Co-hybridization patterns of male-derived DNA labelled with FITC and female-derived DNA labelled with Cy3 were compared between male and female metaphase spreads (Supplementary Fig. 4). Male- and female-derived probes were hybridized with similar intensities to whole chromosomal regions, including the X and Y chromosomes of males and females in these species. As a result, no male- and female-specific regions were unambiguously detected in this analysis.

**Discussion**
The long-term infeasibility of high-fidelity shark cell culture was thought to be attributable to insufficient adaptation of the culture

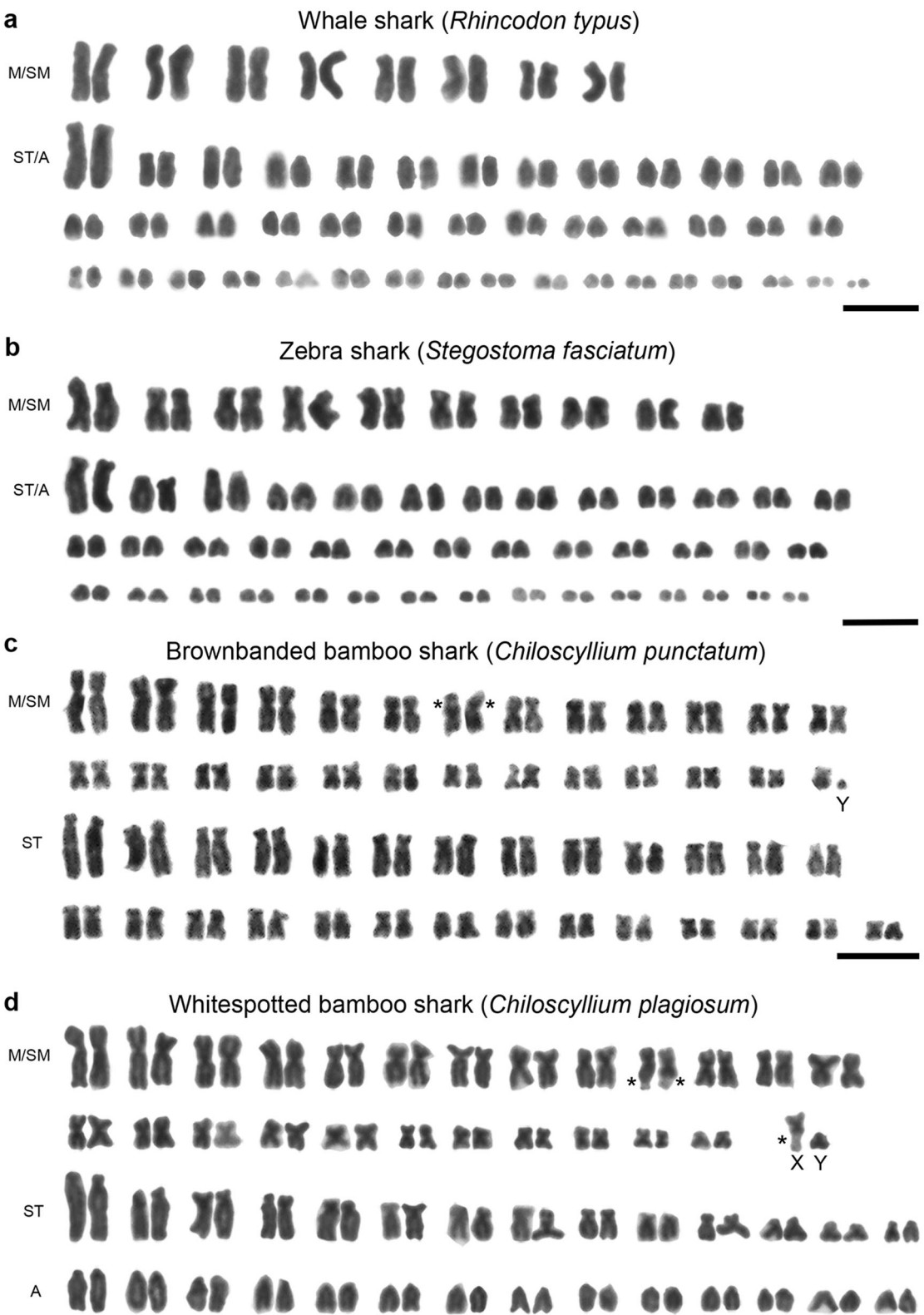

**Fig. 3 Giemsa-stained karyotypes. a** Karyotype of a male of the whale shark *Rhincodon typus* (2*n* = 102). **b** Karyotype of a male of the zebra shark *Stegostoma fasciatum* (2*n* = 102). **c** Karyotype of a male of the brownbanded bamboo shark *Chiloscyllium punctatum* (2*n* = 106). **d** Karyotype of a male of the whitespotted bamboo shark *C. plagiosum* (2*n* = 106). Asterisks indicate the positions of secondary constrictions. M metacentric chromosomes, SM submetacentric chromosomes, ST subtelocentric chromosomes, A acrocentric chromosomes. Scale bars represent 10 µm. See Supplementary Fig. 2 for female karyotypes and Supplementary Fig. 3 for metaphase spreads.

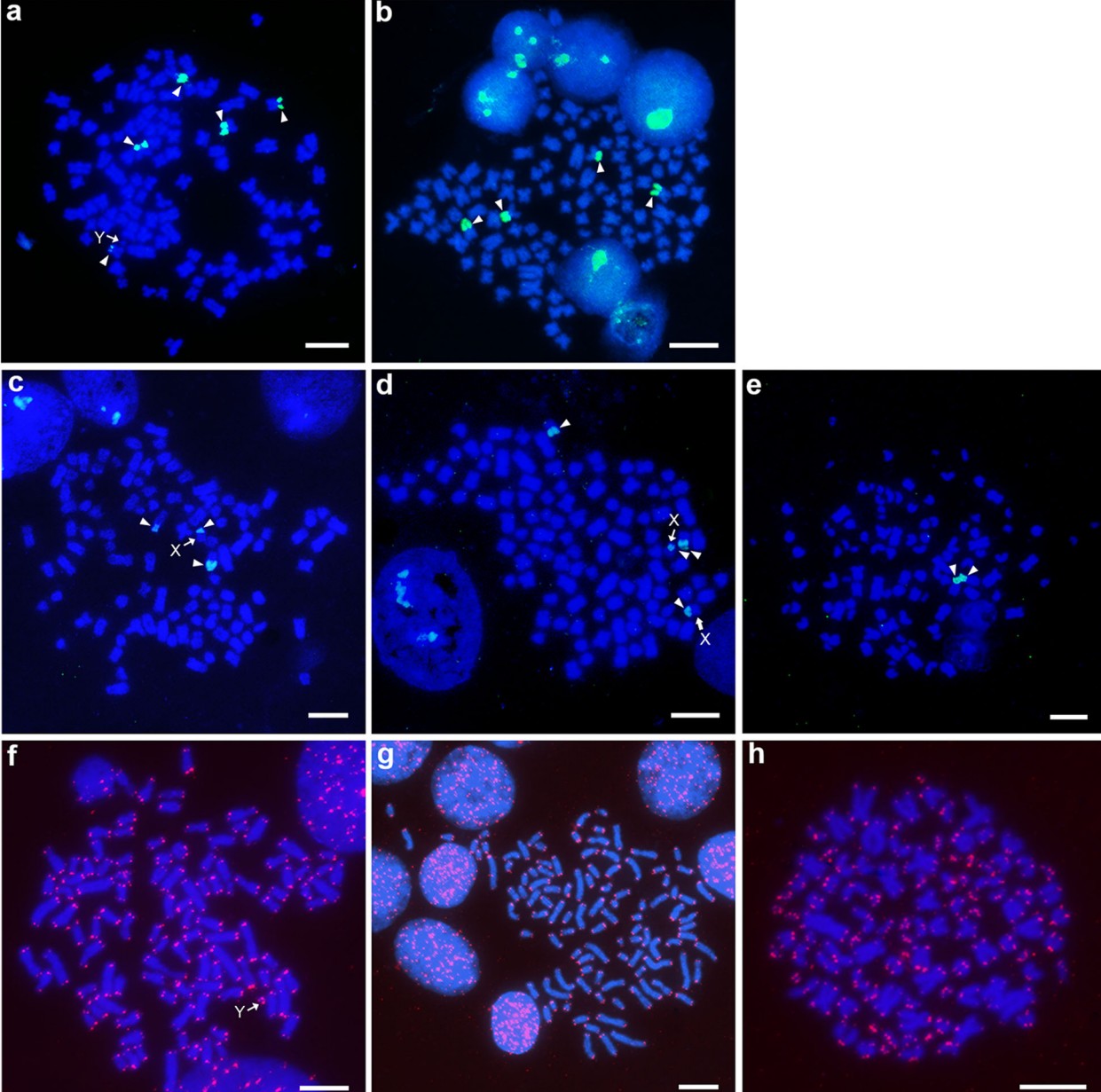

**Fig. 4 Mapping of 18S–28S rDNA and telomeres.** FISH signals of the 18S–28S rRNA genes (arrowheads) are shown for chromosomes prepared from a male (**a**) and a female (**b**) of the brownbanded bamboo shark, a male (**c**) and a female (**d**) of the whitespotted bamboo shark and a female of the zebra shark (**e**). FISH signals of telomeric repeats (red) are shown for chromosomes prepared from a male of the brownbanded bamboo shark (**f**), a male of the whitespotted bamboo shark (**g**) and a female of the zebra shark (**h**). Arrows indicate putative sex chromosomes. Scale bars represent 10 μm.

medium to body fluid osmolality. To overcome this problem, we referred to past attempts and tried novel combinations of medium ingredients. In this study, we proposed methods for cell culture and karyotype analyses of chondrichthyans. Our protocol allowed us to produce high-frequency proliferating cells and chromosome metaphase spreads from the four shark species in Orectolobiformes. Our additional experiments showed the utility of this protocol with other shark species in the order Carcharhiniformes, the banded houndshark *Triakis scyllium* and cloudy catshark *Scyliorhinus torazame*, which confirmed previously reported karyotypes (Supplementary Fig. 5). These results support the wide applicability of our protocol to more diverse chondrichthyans.

Growth of a fibroblast cell line was reported for the spiny dogfish shark *S. acanthias* using culture media supplemented with cell growth factors[29]. However, in this previous study, the cell culture medium was formulated to have osmolality similar to that of osteichthyan blood, but was not adapted to the body fluid of chondrichthyans because only early embryos before prehatching (water penetration through egg cases), namely before the typical high osmolality is acquired, were employed. Moreover, in that study, the culture medium was supplemented with shark yolk extract, which, to our knowledge, is not widely available[29]. In this respect, our present study added urea, NaCl, trimethylamine *N*-oxide and cell growth factors in the medium, which ensured a high-fidelity cell culture (Fig. 2a, b, Supplementary Fig. 1 and Supplementary Data 2). Our culture protocol can accommodate diverse tissue types including prehatched embryos, as well as juvenile tissues, and produced a marked increase in the chromosome metaphase index (Fig. 2d) compared with that produced using other methods[21,23,25,29]. In addition, our protocol does not

require expensive or inaccessible materials, such as shark yolk extract.

Blood lymphocyte culture has a practical advantage of being less invasive and able to be used for repeated sampling, which is crucial in studying long-lived and/or protected species. Our protocol for lymphocyte culture enabled us to observe proliferation of cells from blood and spleens of the shark species using culture medium with a novel combination of the mitogens, Con A, LPS, PHA, and PMA (Supplementary Fig. 1 and Supplementary Data 2). However, the frequency of proliferating cells and chromosome metaphase spreads varied between the examined individuals, especially in the whale shark and zebra shark, as previously reported[26]. Collectively, our cell culture protocol (Supplementary Fig. 1) will contribute to in vitro analyses, including molecular cytogenetic analyses, of chondrichthyans.

Karyotypes of teleost fishes have been reported for more than 3200 species, most of which have 44–54 chromosomes[15]. In comparison, there are fewer karyotype reports for chondrichthyans (83 species)[5,6,15,16] (Fig. 5 and Supplementary Data 1). To our knowledge, this is the first karyotype report of shark species in the order Orectolobiformes. Our analysis revealed relatively high diploid chromosome numbers of the four orectolobiform shark species (102–106) (Fig. 3). Importantly, the two bamboo shark species we studied were shown to have the highest number of chromosomes ($2n = 106$) among all the chondrichthyan species examined to date (Supplementary Data 1). Several species with more than 100 chromosomes are widely distributed in different elasmobranch orders, namely Orectolobiformes, Heterodontiformes[16,36,37], Hexanchiformes[26,36] and Rajiformes[38] (Supplementary Data 1). These observations suggest that the last common ancestor of extant elasmobranchs had a large number (e.g., more than 100) of chromosomes, which decreased independently in the other elasmobranch lineages[5,16]. The holocephalan species in the other chondrichthyan lineage have similar numbers of or fewer chromosomes than elasmobranchs: $2n = 58$ for the spotted ratfish *Hydrolagus colliei*[39] and $2n = 86$ for the rabbit fish *Chimaera monstrosa*[40] (Fig. 5 and Supplementary Data 1). By contrast, comparative genome sequence analyses have suggested a diploid chromosome number of ancestral jawed vertebrates of 80–108 (refs. [41,42]). These sequence-based inferences did not incorporate chondrichthyans for which no chromosome-scale genome sequence information was available. Our present study provides a karyotypic basis for the future organization of awaited genome sequences, which may provide more reliable inference about evolutionary scenarios.

Cartilaginous fishes exhibit remarkable plasticity of their reproductive systems[43]. Some species selected for this study, including the zebra shark, brownbanded bamboo shark and whitespotted bamboo shark, lay eggs (oviparity), whereas others, including the whale shark, give birth to babies (viviparity). Investigation of the sexual differentiation and its genetic trigger is expected to reflect their unique underwater lifestyle and demography. Teleost fishes exhibit extraordinary plasticity of sex determination systems and sex chromosome organization with either environmental sex determination or genetic sex determination (GSD) system[44]. Most of the teleost fish species with the GSD system exhibit male heterogamety (XX/XY) and have less-differentiated sex chromosome pairs than mammals and birds[45]. It is widely thought that the sex of chondrichthyans is also determined by the GSD system with XX/XY sex chromosomes[44,46]. In chondrichthyans, sex chromosomes have been reported for eight species when only those reports based on multiple individuals for both sexes are considered[17,26,47–50] (Fig. 5 and Supplementary Data 1). These species all are myliobatiform or rhinopristiform species and represent only a small subset of the

entire chondrichthyan diversity. It is therefore unclear whether chondrichthyans generally exhibit male heterogamety.

Although not included in the eight species whose sex chromosomes were identified, sex chromosomes have been suggested for 13 more elasmobranch species; however, the studies that have documented this have included only one sex or one individual[17,26,46,51–53] (Supplementary Data 1). To avoid such unreliability, the present study included both sexes and multiple individuals per sex of the brownbanded bamboo shark and whitespotted bamboo shark and revealed differentiated X and Y sex chromosomes in these two species (Fig. 3c, d). In the CGH for these two species, we detected no sex-specific chromosomal regions (Supplementary Fig. 4), which suggests no accumulation of repetitive sequences specific to the Y chromosomes. It is possible that the limited resolution of this CGH method (over megabases) did not allow the detection of Y-specific repetitive sequences. Consequently, these data suggest that the Y chromosomes in these two species are in the middle of an evolutionary transition of sex chromosome differentiation. Further cytogenetic and genomic analyses will enable us to understand more about the evolution of karyotypes including sex chromosomes in Chondrichthyes.

Considering all existing information and our data, we conclude that chondrichthyan karyotypes are generally characterized by numerous chromosomes (up to 106), with a large distribution of chromosome lengths in a karyotype, and sometimes include heteromorphic sex chromosomes. Most chondrichthyan species remain to be analysed, but this emerging karyotypic format is distinct from that of teleost fishes with relatively constant numbers (44–54) of chromosomes that often include homomorphic sex chromosomes[15,45]. The karyotype provides information about a species' inherent chromosome number and sizes, as well as centromere positions, and provides the ultimate goal of whole-genome sequence reconstruction. Recent technical advances using proximity-guided assembly, such as Hi-C, have revealed chromosome-long genome sequences[54]. However, the lack of karyotype reports for many chondrichthyan species has hindered the validation of the product of chromosome-scale genome assembly. Our study has paved the way for more controllable genome analysis of cartilaginous fishes of the current standard.

## Methods

**Animals**. We obtained blood samples from four juveniles of the whale shark *Rhincodon typus* and six adults of the zebra shark *Stegostoma fasciatum*, eight whole embryos and blood samples of six adults of the brownbanded bamboo shark *Chiloscyllium punctatum* and five whole embryos and tissues from one juvenile of the whitespotted bamboo shark *C. plagiosum* (Table 1). The whole blood of the whale shark, zebra shark and brownbanded bamboo shark was obtained from captive animals at the Osaka Aquarium Kaiyukan (a 4.20-m-long male and a 6.05-m-long female of the whale shark and three males and three females of the brownbanded bamboo shark) and the Okinawa Churaumi Aquarium (an 8.68-m-long male and an 8.04-m-long female of the whale shark and three males and three females of the zebra shark). Sampling at these aquariums was conducted by veterinary staff in accordance with the Husbandry Guidelines approved by the Ethics and Welfare Committee of Japanese Association of Zoos and Aquariums[55]. Fertilized eggs of the brownbanded bamboo shark and whitespotted bamboo shark were obtained from Osaka Aquarium Kaiyukan and Suma Aqualife Park in Kobe, respectively. After transfer to aquarium tanks at the RIKEN Kobe Campus, the bamboo shark eggs were cultured at 25 °C in artificial seawater until embryonic developmental stage 32–34 after initiation of male clasper development, according to the staging table released previously[56]. A 45-cm-long male juvenile of the whitespotted bamboo shark was purchased from a commercial marine organism supplier in Izunokuni city, Shizuoka Prefecture, Japan, in March 2019. All other experiments were conducted in accordance with the institutional guideline Regulations for the Animal Experiments and approved by the Institutional Animal Care and Use Committee of RIKEN Kobe Branch.

**DNA-based species identification**. For molecular identification of the brownbanded bamboo shark and whitespotted bamboo shark, genomic DNA was extracted from the tails of embryos and livers of a juvenile using a DNeasy Blood & Tissue Kit (Qiagen, Hilden, Germany). Partial sequences of *COX1* in the

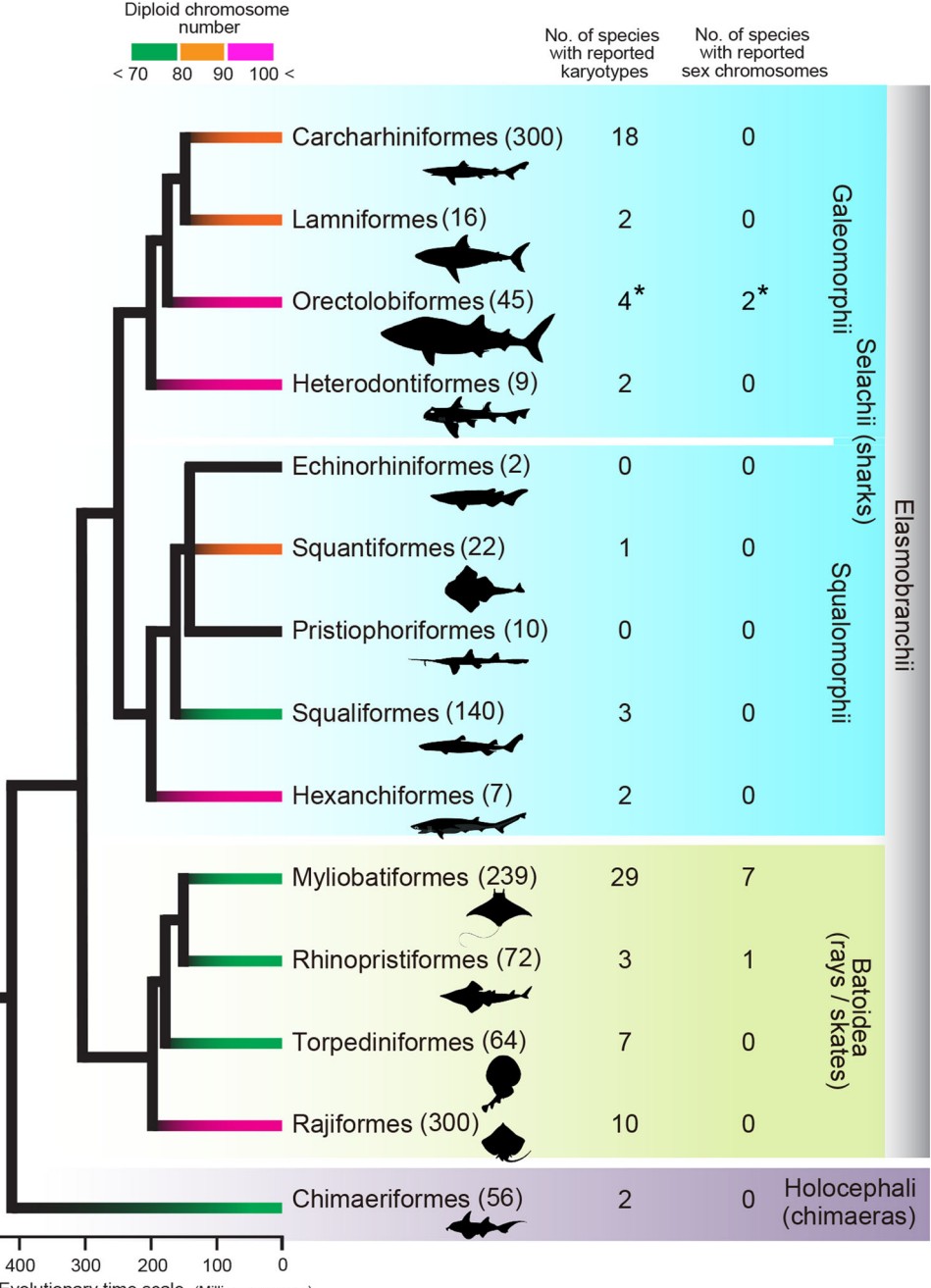

**Fig. 5 Summary of chondrichthyan karyotype studies.** The karyotype information is detailed in Supplementary Data 1. The numbers of species in individual orders are shown in the parentheses based on an existing resource[10]. Asterisks indicate the numbers of species whose karyotypes and sex chromosomes were identified in the present study. The phylogenetic tree and divergence times are based on the existing literature[60-63].

mitochondrial genome were amplified using PCR and then determined. The forward primer 5′-GCATGAGCAGGAATRGTAGGT-3′ and the reverse primer 5′-ATCAACTGATGCTCCTGCRT-3′ were designed based on the nucleotide sequences of the following species in the genus *Chiloscyllium*: *C. punctatum* (NCBI GenBank Accession ID: JN313263), *C. plagiosum* (MG574425), *C. burmensis* (MH429291), *C. hasseltii* (MH235620) and *C. indicum* (MG644344). Nucleotide sequences of the DNA fragments labelled with a BigDye Terminator v3.1 Cycle Sequencing Kit (Thermo Fisher Scientific-Applied Biosystems, Carlsbad, CA, USA) were determined using an Applied Biosystems 3730xl DNA Analyzer (Thermo Fisher Scientific-Applied Biosystems).

**Fibroblast cell culture.** Whole embryos and juvenile tissues (kidneys and peritoneum) were used after cooling with crushed ice for anesthetization. The whole embryos and tissues were washed in Dulbecco's modified Eagle's medium (Thermo Fisher Scientific-GIBCO, Carlsbad, CA, USA) containing a high concentration

(5%) of antibiotic–antimycotic solution (Thermo Fisher Scientific-GIBCO), 373 mM urea and 89 mM NaCl with the pH adjusted to 7.3 (ref. [26]). The washed whole embryos and tissues were minced with sterilized scissors and plated on a collagen I-coated culture dish (AGC Techno Glass, Shizuoka, Japan), and cultured in LDF medium, a mixture of 50% Dulbecco's modified Eagle's medium, 35% L-15 and 15% Ham's F-12, supplemented with 12% foetal bovine serum (FBS), 1% antibiotic–antimycotic solution, 1% ITS-G, 100 μg/ml kanamycin, 2 ng/ml epidermal growth factor, 2 ng/ml fibroblast growth factor (all from Thermo Fisher Scientific-GIBCO), 333 mM urea, 188 mM NaCl and 54 mM trimethylamine *N*-oxide, with the pH adjusted to 7.3 (refs. [22,23,25,29,31]). The cultures were incubated at 26 °C in a humidified atmosphere of 5% $CO_2$. Primary cultured fibroblasts were harvested using 1.46 U/ml Dispase II (Thermo Fisher Scientific-GIBCO) in shark PBS[26], which is conventional PBS supplemented with 299 mM urea and 68 mM NaCl, and then subcultured no more than seven times to avoid chromosomal aberrations.

**Primary lymphocyte culture**. Heparinized blood (2–4 ml) was thoroughly mixed with 6 ml of cold wash medium, RPMI 1640 medium supplemented with 12% FBS, 1% antibiotic–antimycotic solution (all from Thermo Fisher Scientific-GIBCO), 373 mM urea and 89 mM NaCl, with the pH adjusted to 7.3, in a 15-ml sterile plastic tube, placed on ice for 5 min and centrifuged at 1000 r.p.m. for 7 min at room temperature. After centrifugation, the buffy coat, which contains lymphocytes, was floated in plasma by a gentle stirring with a pipette (stirring method)[35]. The lymphocytes were suspended in RPMI 1640 medium supplemented with 12% FBS, 1% antibiotic–antimycotic solution, 0.5% ITS-G (all from Thermo Fisher Scientific-GIBCO), 25 µM mercaptoethanol, 373 mM urea, 89 mM NaCl, and mitogens such as 15 µg/ml Con A (type IV-S) (Sigma-Aldrich, St. Louis, MO, USA), 100 µg/ml LPS (Sigma-Aldrich), 18 µg/ml PHA (HA15) (Thermo Fisher Scientific, Carlsbad, CA, USA) and 15 µg/ml PMA (Sigma-Aldrich), with the pH adjusted to 7.3 (refs. [24,26,35]). For collection of lymphocytes from a spleen, the spleen was crushed between two sterilized glass slides in 6 ml of wash medium. After centrifugation at 1000 r.p.m. for 5 min at room temperature, the lymphocytes were suspended in the culture medium used above. The lymphocytes from blood and spleen were cultured in plastic bottles for 2–6 days at 26 °C in a humidified atmosphere of 5% $CO_2$.

**Chromosome preparation**. Following harvesting, the cultured fibroblasts and lymphocytes were collected after colcemid treatment (150 ng/ml) for 1–3 h, subjected to hypotonic treatment in 0.075 M KCl for 20–40 min and fixed in methanol/acetic acid (3:1). The cell suspension was dropped onto a glass slide and air-dried, and the slides were kept at −80 °C until use. For karyotyping, the slides were stained with 3% Giemsa solution (pH 6.8) for 10 min. The methods from cell culture to chromosome preparation are charted in Supplementary Fig. 1.

**Fluorescence in situ hybridization**. To determine the chromosomal location of the 18S–28S rRNA genes, we used pHr21Ab (5.8 kb for the 5′ portion) and pHr14E3 (7.3 kb for the 3′ portion) fragments of the human 45S pre-ribosomal RNA gene (*RNA45S*), which encodes a precursor RNA for 18S, 5.8S and 28S rRNAs, as the FISH probe as in our previous studies[19,20,31]. The DNA fragments, which were provided by National Institutes of Biomedical Innovation, Health and Nutrition, Osaka, were labelled with biotin 16-dUTP using a nick translation kit (Roche Diagnostics, Basel, Switzerland) following the manufacturer's instructions, and ethanol precipitated with salmon sperm DNA and *Escherichia coli* RNA (all from Sigma-Aldrich), followed by denaturation at 75 °C for 10 min in 100% formamide[31,57]. The chromosome slides were hardened at 65 °C for 2 h, denatured at 70 °C for 2 min in 70% formamide/2× SSC, and dehydrated in 70 and 100% ethanol at 4 °C for 5 min each. A mixture containing the denatured DNA, 50% formamide/2× SSC, 10% dextran sulfate, and 2 µg/µl BSA was put on the denatured chromosome slides and covered with parafilm, and the slides were incubated overnight at 37 °C. After hybridization, the slides were washed for 20 min in 50% formamide, 2× SSC at 37 °C, and in 2× SSC and 1× SSC for 15 min each at room temperature. After rinsing the slides in 4× SSC for 5 min, the slides were incubated under parafilm with avidin, Alexa Fluor 488 conjugate (Thermo Fisher Scientific-Molecular Probes, Carlsbad, CA, USA) at a 1:500 diluted in 1% BSA, 4× SSC for 1 h at 37 °C. The slides were washed on the shaker with 4× SSC, 0.1% Nonidet P-40/4× SSC, 4× SSC for 10 min each at room temperature. After rinsing the slides in 2× SSC for 5 min, the slides were mounted with Vectashield mount medium with DAPI (Vector Laboratories, Burlingame, CA, USA). For chromosomal mapping of telomeres, DIG-labelled 42-bp-long oligonucleotide sequences, $(TTAGGG)_7$ and $(TAACCC)_7$, were used, and the probe was stained with rhodamine-conjugated anti-digoxigenin Fab fragments (Roche Diagnostics).

**Comparative genomic hybridization**. CGH was performed using the method of FISH with slight modification[58,59]. We used genomic DNA of one individual per sex among the genomic DNAs used in the analysis for DNA-based species identification of the brownbanded bamboo shark and whitespotted bamboo shark. Female and male genomic DNA was labelled with FITC-dUTP (Thermo Fisher Scientific-Molecular Probes) and CyDye3-dUTP (GE Healthcare, Buckinghamshire, UK), respectively, using a nick translation kit (Roche Diagnostics), and ethanol precipitated with salmon sperm DNA and *E. coli* tRNA (all from Sigma-Aldrich), followed by denaturation at 75 °C for 10 min in 100% formamide. A mixture containing the denatured DNA, 50% formamide, 2× SSC, 10% dextran sulfate, and 2 µg/µl BSA was put on the denatured chromosome slides and covered with parafilm, and the slides were incubated at 37 °C for 3 days. After hybridization, the slides were washed in 4× SSC, 0.1% Nonidet P-40/4× SSC, 4× SSC and 2× SSC for 5 min each at room temperature and mounted with Vectashield mount medium with DAPI (Vector Laboratories).

**Reporting summary**. Further information on research design is available in the Nature Research Reporting Summary linked to this article.

## Data availability

The *COX1* genes in the brownbanded bamboo shark and whitespotted bamboo shark, which we used for species identification of these two *Chiloscyllium* species, have been deposited in the GenBank database (brownbanded bamboo shark: LC537758 and whitespotted bamboo shark: LC537759). All data supporting the findings of this study are available within the Article and its Supplementary Information and/or from the corresponding author upon reasonable request.

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

## Acknowledgements

We thank Rui Matsumoto at the Okinawa Churaumi Aquarium and Kiyonori Nishida, Takaomi Ito and Yuka Inata at the Osaka Aquarium Kaiyukan for blood sampling; other staff at the Osaka Aquarium Kaiyukan and Daiki Senda for helping with egg collection; and Koh Onimaru for helping with the developmental staging of bamboo shark embryos. Our gratitude extends to Takashi Asahida for insightful discussion of elasmobranch karyotypes. This work was supported in part by JSPS KAKENHI Grant No. 17K07511 to Y.U., a research grant from MEXT to the RIKEN Center for Biosystems Dynamics Research, and intramural grants within RIKEN including the All-RIKEN "Epigenome Manipulation Project" to S.K.

## Author contributions

Y.U. and S.K. conceived the study. R.N., I.K., N.H., S.S., K.M., and K.S. provided samples. Y.U. performed the experiments. Y.U. and S.K. interpreted the data and drafted the manuscript. All authors contributed to the final manuscript editing.

## Competing interests

The authors declare no competing interests.
