## [Peer Review File · Communications Biology]

Reviewers' comments:

Reviewer #1 (Remarks to the Author):

To preserve formatting I have attached an identical copy of these comments to the review, but I have pasted the comments here as well.

In this paper the authors describe a novel method for cell culture in chondrichthyans which historically has been a difficult group for cell culture and karyotype determination. The culturing method was used to create chromosome preparations for determining the karyotypes of four shark species from the order Orectolobiformes: the whale shark (*Rhincodon typus*), the zebra shark (*Stegostoma fasciatum*), the brownbanded bamboo shark (*Chiloscyllium punctatum*), and the whitespotted bamboo shark (*Chiloscyllium plagiosum*). For each of these four species the authors determined the chromosome numbers and types using both male and female individuals and determined the diploid chromosome numbers to be 102, 102, 106, and 106 for the species respectively. In addition, the authors conducted FISH analyses to identify 18S-28S rRNA gene locations on several chromosomes for each species and telomeric sequences on all chromosomes. Using evidence of size differences in chromosomes present in male and female preparations, the authors also identified putative sex chromosomes for the two bamboo shark species and an XX/XY system. Comparative genomic hybridization failed to identify sex specific regions leading the authors to conclude that sex chromosome differentiation is relatively new in these species.

Overall, my impression is that the findings and use of cell culture to obtain karyotype information for members of the Chondrichthyes is a very significant advance for that evolutionarily key class. As the authors point out, karyotype information in this taxonomic group is woefully understudied compared to other taxonomic groups, and it is important to rectify this to provide context and a final goal for genome sequencing projects. However, the goal/scope of the paper is somewhat disconnected when considering the introduction (which focuses on karyotyping) and the discussion (which focuses on sex determination systems). Additionally, without further taxonomic sampling it is unclear of whether the methods described will be of the same utility in the other orders of the Chondrichthyes. In regards to the discussion of sex determination in bamboo sharks, though it is thought that most Chondrichthyans do have an XX/XY system, the discussion ignores the stunning diversity of systems seen in other fish. The availability of genomic sequencing data for some of the species that they studied could be utilized for identifying candidate sequences and expanding their validation of sex chromosome identification from the karyotype work. In short, the study would benefit from focusing and expanding on the discussion of karyotyping from cell culture in additional species (or by expanding their discussion of the relationship between their data and previously hypothesized evolution of genome and chromosome numbers in the literature that is cited) or from expanding on the work on sex determination in bamboo sharks. Below I have some specific comments and suggestions. I am not an expert on cell culture and will leave any comments on that aspect to my fellow reviewers, but the details to that effect and solutions proposed by the authors (particularly the use of a combination of mitogens) seem informative and significant.

1. Figure 1 seems to add relatively little to the manuscript as currently constructed. It would have greater utility as a panel within the image used for figure 5, displaying where the 4 species are found within the chondrichthyan tree. Additionally, it needs a scale bar to indicate the phylogenetic distance represented by the branches in the tree displayed. I would suggest shifting this figure to a panel and instead include a table of the species in the studied, the number, sex, and life stage of individuals used. Currently that information is buried at the end of the manuscript in the methods when the information should appear earlier within the results...a table would accomplish this more quickly.

2. If the focus of the paper is to be on the cell culture techniques and advances in the paper then it would be appropriate to expand the discussion of the challenges of cell culture beyond the

mention that the blood osmolality is greater in chondrichthyans (lines 48-49, 78-80). What particular types of tissue culture are impacted by this and how have efforts compared to some of the marine teleosts that have osmolality that is at least closer to that of chondrichthyans.

3. Lines 57-59, A minor/wording comment is that the way that the sentences here read seem to highlight that the whale shark differs from other Orectolobiformes in that Orectolobiformes have a temperate/tropical distribution in the Pacific Ocean and that the whale shark does not. I assume that the authors rather mean to highlight what is stated next, that the whale shark is the world's largest extant fish and that this makes it different from the rest of the Orectolobiformes (since its distribution also includes the Pacific). Please just adjust the wording to make this clear.

4. There is little to no mention of genetic sex determination (gsd) in the introduction of the paper despite it being a major focus of the results and discussion. I would suggest adding some text discussing about gsd in Chondrichthyes and the variety of mechanisms seen in teleost fish to the introduction. In that way the authors could present to the reader what the reader should expect to see as evidence of gsd in the karyotype data.

5. In the first section of the results I would present what and how many of each species are sampled in the study. It is not until the 3rd section of the results that it is made clear what exact species are sampled. My suggested resolution of my first comment (changing figure 1 to a table of species and numbers sampled) would address this.

6. A minor comment, on line 166 please add the word 'only' to indicate that signal was seen on only four chromosomes rather than signal being present on four or more of the chromosomes.

7. I would suggest adding the word 'putative' to the description of the Y chromosomes in line 170-171. None of the chromosomes in figure 4c have been conclusively been determined to be the Y chromosome so their classification should be qualified.

8. In the section starting on line 182, the acronyms used should be defined. This is done for prior sections but here the definition of the acronyms such as CGH does not occur until much later in the manuscript. It should be done here at first mention.

9. Within the discussion, I would like to see more of a conversation of the implication of the results to hypotheses put forth regarding chromosome evolution and genome size that were put forth in some of the studies by Schwartz, Maddock, Stingo, Rocco, and other authors previously cited in this paper that looked at cytogenetics in sharks. This is done slightly in the paragraph from lines 222-235 but I think that an expansion and a discussion of what groups in the Chondrichthyes should be targeted with their methods would strengthen the conclusion on lines 233-235.

10. I would suggest condensing the discussion of reproductive biology from lines 236-242. The data in this paper don't really have the power to address some of these aspects, though the mode of sex determination in species with those characteristics would be useful and so a brief mention that chondrichthyans have a diverse and remarkable array of reproductive characteristics is reasonable.

11. Line 244, the wording of this sentence (when considering the wording of the next sentence as well) makes it seem that fish mainly exhibit an xx/xy system of gsd, but other modes of sex determination are known in fish and I would suggest rewording this to reflect that while the system is seen, fish have other systems as well.

12. It is a minor comment but the focus on the whale shark in the wording of the final sentence of the discussion seems unnecessary as the analyses performed on whale shark were similar to those on other species in the paper. If any species seems to be deserve greater focus in the discussion it is the two bamboo shark species that are focused on for the FISH and CGH analyses.

13. In line 312 and the section that it is found in, the authors should include details regarding the type of cloning performed and sequencing done to determine COI from the bamboo sharks alone with criteria used for species identification (% mismatch allowed). I assume Sanger was used rather than an amplicon method but I am not sure what was done from reading it.

14. Line 321, is cashed a typo that should be changed to crushed?

15. More detail is required for the methods on comparative genome hybridization on lines 391-394. The cited papers are work done in anurans and it is unclear what specific differences there may have been in this paper or the source of male or female DNA for the hybridizations. Were the same individuals for chromosome spreads used or were additional males and females sampled?

16. For references 8 and 52, access dates need to be added since these sources are databases that are frequently updated with new information.

Reviewer #2 (Remarks to the Author):

The authors successfully optimized cell culture conditions for shark fibroblasts and lymphocytes, which is a difficult task because the conditions must mimic the peculiar features of body fluid osmolality of marine chondrichthyans. Thereafter, they karyotyped and characterized for NORs, telomeric sequences and sex chromosomes 4 new shark species. The study is an essential contribution to chromosome-scale genome assemblies for these species.

The manuscript is written clear and the results and their interpretations are straightforward. Though, the text could use additional editing for English. I have just a few minor comments and suggestions.

Comments

1. Title: please replace "as a ground". I recommend "as an important asset", "as a resource...", "as an important foundation", etc.
2. One of the important achievements of this study was optimizing cell culture condition for sharks. This is not sufficiently detailed and highlighted in Results. Please elaborate.
3. The symbol for cytochrome oxidase subunit 1 should be COX1 not COI.
4. Page 5, "CGH patterns...": CGH must be explained at its first appearance and it is "Comparative Genomic Hybridization" not "...Genome...".
5. Page 7, line 3: should be "...relatively high diploid chromosome number..", not "large"
6. Page 7, line 17: should be "hermaphroditism" not "hermaphrodism"
7. Page 7, line 33: should be "differentiated X and Y sex chromosomes".
8. Page 8, lines 9-11: the sentence "A karyotype instructs the species' inherent chromosome number and sizes, as well as centromere positions and provides an ultimate goal of whole-genome sequencing." reads odd. Please reword.
9. Fig. 1 needs a note whether these photos are taken by the authors or retrieved from internet. The latter probably requires permission or at least a webaddress.
10. Supp Fig. 1: should be "supernatant" not "supermatant".

Reviewer #3 (Remarks to the Author):

Karyotyping in elasmobranch species has been problematic because of the difficulty in producing high quality mitotic preparations. The use of cell culture to generate better chromosome

preparations has been investigated by this group and their use of greatly modified culture media specific for elasmobranch physiology seems to be the key. In essence, this paper is really a methods paper that describes the modified media and the culture techniques. The chromosomes generated seem fine, albeit a bit contracted in some cases. The use of the chromosomes for CGH and in situ hybridization is good evidence that the chromosome preparations are of good quality, certainly better than a lot of the descriptions in the literature that have used direct tissue preparations.

All-in-all, this manuscript is very descriptive. The advancement here is the use of modified culture media; however, this in-and-of-itself is but a small advancement because alterations in culture conditions are necessary for any organism for which physiological conditions are far from the norm.

Given the above and the lack of major findings that will move the field forward significantly, this paper seems inappropriate for the broad readership of this journal. Rather, it seems to be better suited to a journal focusing on shark biology or cytogenetics or cell culture.

minor points:

There are many typographical and grammatical errors in the text.

The authors refer to epidermal cells but probably intended to say epithelial cells.

While the karyograms are somewhat informative, the authors ought to show the corresponding giemsa metaphase spreads as well to show the overall quality of the spreads.

As effective as the modified culture media is for generating dividing cells for chromosome preparation, it is cost-prohibitive for many labs who may want to routinely karyotype chondrichthyan species. Has the group empirically tested different conditions and formulations? How did they come up with this formulation to begin with.

Overall report of our revision

This letter reports our revision and describes our point-by-point responses (in the indented lines) to the reviewers' comments and is followed by a manuscript text highlighting the individual changes from the initially submitted version. Revisions have been highlighted with track changes in the manuscript. Apart from our point-by-point responses to individual reviewers' comments, we have modified a part of the manuscript as follows.

First, as a response to the reviewers' suggestions, we have had our revised manuscript checked by professional proofreaders. Second, because the exact names of the two *Chiloscyllium* species did not appear early enough in the initially submitted manuscript, we have included the species names together with the common names of the four shark species we used, in the last paragraph of the **Introduction**. Third, although not directly related to the contents of our findings, we have included an alias of the zebra shark, *Stegostoma tigrinum*, at the first appearance of this species in the manuscript (again, in the last paragraph of the **Introduction**), based on a recent proposal of this species name as the senior synonym (Dahl et al. Copeia 107:524-). Still, we have maintained the long-standing species name *Stegostoma fasciatum*, because it remains in traditional resources such as the NCBI Taxonomy database.

A part of the last paragraph of the **Introduction**:

'In this study, we have derived protocols for cell culture of fibroblasts using multipassage culture and optimized the cell culture protocols of lymphocytes for four orectolobiform shark species including, the whale shark, zebra shark Stegostoma fasciatum (also known as S. tigrinum³³), brownbanded bamboo shark Chiloscyllium punctatum and whitespotted bamboo shark C. plagiolum (Fig. 1).

Response to Reviewer #1:

Overall, my impression is that the findings and use of cell culture to obtain karyotype information for members of the Chondrichthyes is a very significant advance for that evolutionarily key class. As the authors point out, karyotype information in this taxonomic group is woefully understudied compared to other taxonomic groups, and it is important to rectify this to provide context and a final goal for genome sequencing projects. However, the goal/scope of the paper is somewhat disconnected when considering the introduction (which focuses on karyotyping) and the discussion (which focuses on sex determination systems).

Thank you for pointing this out. We admit that the description of karyotyping versus sex-associated themes was not consistently balanced in the originally submitted manuscript. In the revised manuscript, we have made our focus on karyotyping methods and results clearer, which is outlined in the following responses to the individual comments and suggestions.

Additionally, without further taxonomic sampling it is unclear of whether the methods described will be of the same utility in the other orders of the Chondrichthyes.

This point is addressed by our success in other shark species, cloudy catshark *Scyliorhinus torazame* and banded hound shark *Triakis scyllium*, belonging to a different order Carcharhiniformes. Because the karyotypes of these species had been previously reported by other groups (Asahida, et al. 1988 Jpn. J. Ichthyol.35: 215-, Asahida & Ida 1989 Jpn. J. Ichthyol. 36: 275-), we did not include any data for these species in the initially submitted manuscript. To show a wide utility of our protocol, we have newly included metaphase chromosome spreads for these species in **Supplementary Fig. 5** and mention this in **Discussion**. Although we have not tested, we think that our protocol is applicable to species in the other shark orders because of similar osmolality among chondrichthyans (ref. 19: Griffith 1981).

In Discussion:

'Our protocol allowed us to produce high-frequency proliferating cells and chromosome metaphase spreads from the four shark species in Orectolobiformes. Our additional experiments showed the utility of this protocol with other shark species in the order Carcharhiniformes, the banded houndshark *Triakis scyllium* and cloudy catshark *Scyliorhinus torazame*, which confirmed previously reported karyotypes (Supplementary Fig. 5). These results support the wide applicability of our protocol to more diverse chondrichthyans.'

In short, the study would benefit from focusing and expanding on the discussion of karyotyping from cell culture in additional species (or by expanding their discussion of the relationship between their data and previously hypothesized evolution of genome and chromosome numbers in the literature that is cited) or from expanding on the work on sex determination in bamboo sharks.

As mentioned above in our response to an earlier suggestion, we have included the data for two additional species in the order Carcharhiniformes in **Supplementary Fig. 5** and mention this in **Discussion**.

We understand that genome evolution should be scrutinized by comparing genome sequences rather than just chromosome numbers. Our present study provides the indispensable cytogenetic basis for a new study phase with chromosome-scale genome

sequences. Although we are in a standpoint to refrain from discussing genome evolution only from increase/decrease of chromosome numbers, we have inserted the sentences below with existing information of holocephalan karyotypes and inferred ancestral jawed vertebrate karyotypes, acknowledging those valuable cytogenetic studies.

'The holocephalan species in the other chondrichthyan lineage have similar numbers of or fewer chromosomes than elasmobranchs: $2n = 58$ for the spotted ratfish *Hydrolagus collicii*³⁷ and $2n = 86$ for the rabbit fish *Chimaera monstrosa*³⁸ (Fig. 5, Supplementary Table 1). By contrast, comparative genome sequence analyses have suggested a diploid chromosome number of ancestral jawed vertebrates of 80–108^{39,40}. These sequence-based inferences did not incorporate chondrichthyans for which no chromosome-scale genome sequence information was available. Our present study provides a karyotypic basis for the future organization of awaited genome sequences, which may provide more reliable inference about evolutionary scenarios.'

New citations:

37. Ohno, S. *et al.* Microchromosomes in holocephalian, chondrosteian and holostean fishes. *Chromosoma* **26**, 35–40 (1969).
38. Nygren, A. & Jahnke, M. Microchromosomes in primitive fishes. *Swed. J. Agric. Res.* **2**, 229–238 (1972).
39. Nakatani, Y., Takeda, H., Kohara, Y. & Morishita, S. Reconstruction of the vertebrate ancestral genome reveals dynamic genome reorganization in early vertebrates. *Genome Res.* **17**, 1254–1265 (2007).
40. Sacerdot, C., Louis, A., Bon, C., Berthelot, C. & Roest Crolius, H. Chromosome evolution at the origin of the ancestral vertebrate genome. *Genome Biol.* **19**, 166 (2018).

Regarding sex determination of sharks, we are conducting an investigation to tackle this, which requires totally different materials (e.g., tissues of developing gonads) and technical solutions (e.g., high-quality genome assembly, genome-wide male/female association study, and functional validation of sex determining gene candidates). Even if we succeed in unveiling the mechanism, we will be publishing the results in a separate manuscript in the future.

1. Figure 1 seems to add relatively little to the manuscript as currently constructed. It would have greater utility as a panel within the image used for figure 5, displaying where the 4 species are found within the chondrichthyan tree. Additionally, it needs a scale bar to indicate the phylogenetic distance represented by the branches in the tree displayed. I would suggest shifting this figure to a panel and instead include a table of the species in the studied, the number, sex, and life stage of individuals used. Currently that information is buried at the end of the manuscript in the methods when the

information should appear earlier within the results...a table would accomplish this more quickly.

We agree that the readers will not be fully instructed with the branch lengths of the tree in **Fig. 1**. In the revised manuscript, we have placed the four species in this tree with the branches of the same length. Apart from this, we would like to keep this figure as **Fig. 1** to serve basic information about the animals and their phylogenetic relationship to non-expert of sharks.

Regarding the idea to tabulate our materials, we agree with it and have newly prepared a table of the species studied as well as the number, sex, and life stage of individuals used. We have included this table as **Table 1**.

2. If the focus of the paper is to be on the cell culture techniques and advances in the paper then it would be appropriate to expand the discussion of the challenges of cell culture beyond the mention that the blood osmolality is greater in chondrichthyans (lines 48-49, 78-80). What particular types of tissue culture are impacted by this and how have efforts compared to some of the marine teleosts that have osmolality that is at least closer to that of chondrichthyans.

For cell culture using marine teleost fishes, the same osmolality of culture media with mammals enabled to detect cell proliferation and obtain chromosome metaphase (ref. 34: Fujiwara et al. 2001) because body fluid osmolality of marine teleost fishes is approximately equal to that of mammals and is different from that of any tissues (except early embryos) of chondrichthyans (ref. 19: Griffith 1981) (included in the first paragraph of **Discussion**). Therefore, we have modified and the sentences below in the third last paragraph of **Introduction**.

~~'To adapt~~For example, the culture medium used for marine teleost fishes cannot be readily applied to ~~their~~chondrichthyans. The medium formulations must be optimized because of the high osmolality, ~~the cell culture condition is required to be modified by optimizing media formulations. Some researchers improved for cell culture of fibroblasts and lymphocytes and established fibroblast cell lines in chondrichthyans using different media formulations and additives~~20–25. Nevertheless, ~~in~~from any tissue (except for early embryos) of chondrichthyans, cultured.'

3. Lines 57-59, A minor/wording comment is that the way that the sentences here read seem to highlight that the whale shark differs from other Orectolobiformes in that Orectolobiformes have a temperate/tropical distribution in the Pacific Ocean and that the whale shark does not. I assume that the authors rather mean to highlight what is stated next, that the whale shark is the world's largest extant fish and that this makes it

different from the rest of the Orectolobiformes (since its distribution also includes the Pacific). Please just adjust the wording to make this clear.

As suggested, we have modified the sentence as below in the second paragraph of **Introduction**, to tone down about the specialty of the whale shark.

~~'The remarkable exception to~~ Within this is order, the whale shark Rhincodon typus (Fig. 1); is unique as a pelagic species ~~known as~~ with the largest extant body size as a 'fish' (reviewed in ref. 9).'

4. There is little to no mention of genetic sex determination (gsd) in the introduction of the paper despite it being a major focus of the results and discussion. I would suggest adding some text discussing about gsd in Chondrichthyes and the variety of mechanisms seen in teleost fish to the introduction. In that way the authors could present to the reader what the reader should expect to see as evidence of gsd in the karyotype data.

As a response to an earlier comment, we have consolidated our stance to tailor our manuscript to focus more on cell culture methods and primary results from karyotyping rather than sex-associated themes. Thus, we minimized our statement about sex-associated themes in the **Discussion**. As a reflection of this change, we understand that this comment may not be as valid as before.

Still, the information about the duality of GSD and ESD suits our context, and we thus have modified the relevant part of **Discussion** as below.

~~'Sex chromosome pairs of teleost fishes are less differentiated than those of mammals and birds, and many teleost~~ Teleost fishes exhibit an extraordinary plasticity of sex determination systems and sex chromosome organization, with the duality between environmental sex determination (ESD) and genetic sex determination (GSD) systems⁴³. Most of the teleost fish species with the GSD system exhibit male heterogamety (XX/XY)³⁴, and have less-differentiated sex chromosome pairs than mammals and birds⁴⁴. It ~~has is~~ widely ~~been~~ thought that the sex of chondrichthyans is also ~~have~~ determined by the GSD system with XX/XY sex chromosomes^{43,45}.

5. In the first section of the results I would present what and how many of each species are sampled in the study. It is not until the 3rd section of the results that it is made clear what exact species are sampled. My suggested resolution of my first comment (changing figure 1 to a table of species and numbers sampled) would address this.

As mentioned above in our response to an earlier suggestion, we have newly prepared a table of the species studied as well as the number, sex, and life stage of individuals used and included this table as **Table 1**.

6. A minor comment, on line 166 please add the word 'only' to indicate that signal was seen on only four chromosomes rather than signal being present on four or more of the chromosomes.

We have modified the text as suggested.

7. I would suggest adding the word 'putative' to the description of the Y chromosomes in line 170-171. None of the chromosomes in figure 4c have been conclusively been determined to be the Y chromosome so their classification should be qualified.

We have modified the text as suggested.

8. In the section starting on line 182, the acronyms used should be defined. This is done for prior sections but here the definition of the acronyms such as CGH does not occur until much later in the manuscript. It should be done here at first mention.

We have modified the text in **Results** as suggested.

'~~Comparative genomic hybridization~~-CGH patterns between male and female chromosomes. To identify sex-specific chromosomal regions, we performed comparative genomic hybridization (CGH) ~~was performed with~~ using metaphase chromosomes from two individuals per sex...'

9. Within the discussion, I would like to see more of a conversation of the implication of the results to hypotheses put forth regarding chromosome evolution and genome size that were put forth in some of the studies by Schwartz, Maddock, Stingo, Rocco, and other authors previously cited in this paper that looked at cytogenetics in sharks. This is done slightly in the paragraph from lines 222-235 but I think that an expansion and a discussion of what groups in the Chondrichthyes should be targeted with their methods would strengthen the conclusion on lines 233-235.

As mentioned earlier, we are reluctant to discuss genome evolution solely based on the comparison of chromosome numbers. Still, we have supplemented the paragraph outlining chondrichthyan karyotype diversity with citing early studies by Schwartz, Maddock, Stingo, and Rocco (ref. 4: Stingo & Rocco 2001, ref. 14: Schwartz & Maddock 2002).

10. I would suggest condensing the discussion of reproductive biology from lines 236-242. The data in this paper don't really have the power to address some of these aspects, though the mode of sex determination in species with those characteristics would be useful and so a brief mention that chondrichthyans have a diverse and remarkable array of reproductive characteristics is reasonable.

Agreeing this and earlier suggestions, we have toned down with sex-associated themes in **Discussion**. In the revised manuscript, we have deleted the sentence about placentation, parthenogenesis and hermaphroditism in cartilaginous fishes from this part.

'Cartilaginous fishes exhibit ~~a~~ remarkable plasticity of their reproductive systems⁴². Some species, ~~selected for this study~~, including the zebra shark, ~~the~~ brownbanded bamboo shark, ~~and the~~ whitespotted bamboo shark ~~selected for this study~~, lay eggs (oviparity), ~~while whereas~~ others, including the whale shark, give birth to babies (viviparity). ~~The latter is sometimes accompanied by placentation. Moreover, parthenogenesis and hermaphroditism have been occasionally documented^{32,33}. Investigation ~~on~~ of the sexual differentiation and its genetic trigger is expected to ~~hint at~~ reflect their unique underwater lifestyle and demography.'~~

11. Line 244, the wording of this sentence (when considering the wording of the next sentence as well) makes it seem that fish mainly exhibit an xx/xy system of gsd, but other modes of sex determination are known in fish and I would suggest rewording this to reflect that while the system is seen, fish have other systems as well.

We agree with this suggestion. As we mentioned in the response to an earlier comment, we have added the sentences about the GSD/ESD difference of sex-determining mechanisms in teleost fishes into **Discussion**.

12. It is a minor comment but the focus on the whale shark in the wording of the final sentence of the discussion seems unnecessary as the analyses performed on whale shark were similar to those on other species in the paper. If any species seems to be deserve greater focus in the discussion it is the two bamboo shark species that are focused on for the FISH and CGH analyses.

As suggested, we have deleted this part of the sentence as below.

'Our study has paved the way ~~to~~ for more controllable genome analysis of cartilaginous fishes ~~on~~ of the ~~modern-current~~ standard, ~~starting with four~~

~~orectolobiform shark species including the elusive, long-lived largest fish, the whale shark.~~

13. In line 312 and the section that it is found in, the authors should include details regarding the type of cloning performed and sequencing done to determine COI from the bamboo sharks alone with criteria used for species identification (% mismatch allowed). I assume Sanger was used rather than an amplicon method but I am not sure what was done from reading it.

The DNA sequence of the individuals of *Chiloscyllium punctatum* showed the similarity of >99.4% (difference of no more than 2 bp) to that of this species in NCBI (Accession ID: JN313263), while that of *C. plagiosum* of >99.7% (difference of no more than 1 bp) to its sequence in NCBI (MG574425). On the other hand, the identities of sequences with the other *Chiloscyllium* species used for primer design ranged from 90.6–92.6% (24–29 bp) for *C. punctatum* and from 90.6–93.1% (24–33 bp) for *C. plagiosum* (see **Supporting Figure, Supporting Table** included below). We have modified this part of the sentence in **Results** as below.

~~‘The obtained nucleotide sequences of the 325 bp-long DNA fragment of the brownbanded bamboo shark (NCBI GenBank Accession ID: LC537758) and whitespotted bamboo shark (LC537759) were identical showed 99.4–100% identity (difference of 0–2 nucleotides) to those of the individual-respective species (JN313263 and MG574425) deposited in the NCBI Nucleotide database (JN313263 and MG574425), respectively. These DNA sequences in NCBI exhibited the difference of 24 nucleotides in the selected COX1 region between the two species, which validates provided firm evidence of our species identification-determination.’~~

Supporting Table. Nucleotide-level identities (%) between the newly determined and existing *Chiloscyllium* COX1 sequences.

	C. punctatum	C. punctatum *	C. plagiosum	C. plagiosum *	C. burmensis	C. hasseltii
C. punctatum (JN313263)	-					
C. punctatum (LC537758)*	100	-				
C. plagiosum (MG574425)	92.1	92.1	-			
C. plagiosum (LC537759)*	92.1	92.1	100	-		
C. burmensis (MH429291)	90.6	90.6	90.6	90.6	-	
C. hasseltii (MH235620)	92.6	92.6	93.1	93.1	92.6	-
C. indicum (MG644344)	91.1	91.1	92.6	92.6	89.1	91.6

Asterisks show the nucleotide sequences determined in this study.

Supporting Figure. Alignment of the newly determined and existing *Chiloscyllium* COX1 sequences.

```

      10      20      30      40      50      60      70      80      90     100     110
C. punctatum (JN313263) TAGTAGGTATAGCTCTTAGCCCTTTAATCCGCGCTGAATTAAAGTCAACCTGGATCCCTCTAGGTGACGATCAGATTTATAATGTAATCGTAACAGCCCATGCTTTTGT
C. punctatum (LC537758)* .G.....C.....T.....GC.....C.....T.....
C. plagiosum (MG574425) .G.....C.....T.....GC.....C.....T.....
C. plagiosum (LC537759)* .G.....C.....T.....GC.....C.....T.....
C. burmensis (MH429291) .C.....T..A...T..T...C.....C.....T..C.....C.....
C. hasseltii (MH235620) .C.....T..T...C.....G.....T..C.....T.....C.....
C. indicum (MG644344) .....C.....T.....C.....G.....T.....C.....T.....C.....

      120     130     140     150     160     170     180     190     200     210     220
C. punctatum (JN313263) ATAATTTTCTTTATAGTGTGCGCTGTAATAATTGGTGGATTGGAAATTGACTAGTACCCTAATAATTGGTGCGCCTGATAGCCCTTTCCTCGAATAAATAATATAG
C. punctatum (LC537758)* .....G..T.....G.....G..C.....G.....G.....C..A.....T..C.....
C. plagiosum (MG574425) .....G..T.....G.....G..C.....G.....G.....C..A.....T..C.....
C. plagiosum (LC537759)* .....G..T.....G.....G..C.....G.....G.....C..A.....T..C.....
C. burmensis (MH429291) .....C..A..A..C.....C..T...G...T...C..C..A...C...C...A.....
C. hasseltii (MH235620) .....C..A..A..C.....C..T...G...T...C..C..A...C...C...A.....
C. indicum (MG644344) .....G..A.....A..G.....C.....G.....C..A..C.....

      230     240     250     260     270     280     290     300     310     320
C. punctatum (JN313263) CTTTGGATTACTTCTCCTTCATTCTTATTAAGCTCTGCAGGAGTTGAAGCCGGAGCAGGAACAGGGTGAACGTGTTACCCACCTTTAGCAGGTAATT
C. punctatum (LC537758)* .....C.....T.....CC.....T.....
C. plagiosum (MG574425) .....C.....T.....CC.....T.....
C. plagiosum (LC537759)* .....C.....T.....CC.....T.....
C. burmensis (MH429291) .....C.....T.....C.....C.....C..T.....
C. hasseltii (MH235620) .....C.....T.....C.....T..G.....T.....C.....C.....
C. indicum (MG644344) .....C.....TT.....CC.....G.....T.....C.....C.....C.....

```

Asterisks show the nucleotide sequences determined in this study.

As suggested about our methodology, we have also inserted the following sentence about our DNA sequencing method in **Methods**.

‘For molecular identification of the brownbanded bamboo shark and whitespotted bamboo shark, genomic DNA was extracted from tails of embryos and livers of a juvenile using DNeasy Blood & Tissue Kit (Qiagen, Hilden, Germany). Nucleotide sequences of the DNA fragments labelled with a Big Dye Terminator v3.1 Cycle Sequencing Kit (Thermo Fisher Scientific-Applied Biosystems, Carlsbad, California, USA) were determined using Applied Biosystems 3730xl DNA Analyzer (Thermo Fisher Scientific-Applied Biosystems).’

14. Line 321, is cashed a typo that should be changed to crushed?

Thank you for pointing this out. We have modified the text as suggested.

15. More detail is required for the methods on comparative genome hybridization on lines 391-394. The cited papers are work done in anurans and it is unclear what specific differences there may have been in this paper or the source of male or female DNA for the hybridizations. Were the same individuals for chromosome spreads used or were additional males and females sampled?

To clarify the source of DNA, we have inserted the following sentence on the CGH method.

‘We used genomic DNA of one individual per sex among the genomic DNAs used in the analysis for DNA-based species identification of the brownbanded bamboo shark and whitespotted bamboo shark.’

The experiment procedure (labeling of probes, hybridization to chromosomes, and washing after hybridization) of the present CGH analysis on sharks is the same as the procedure employed in our previous studies on anuran species (ref. 57, 58: Uno Y et al. 2008, 2015).

16. For references 8 and 52, access dates need to be added since these sources are databases that are frequently updated with new information.

Thank you for pointing this out. As suggested, we have included the access date for these online references.

Response to Reviewer #2:

The authors successfully optimized cell culture conditions for shark fibroblasts and lymphocytes, which is a difficult task because the conditions must mimic the peculiar features of body fluid osmolality of marine chondrichthyans. Thereafter, they karyotyped and characterized for NORs, telomeric sequences and sex chromosomes 4 new shark species. The study is an essential contribution to chromosome-scale genome assemblies for these species.

The manuscript is written clear and the results and their interpretations are straightforward. Though, the text could use additional editing for English. I have just a few minor comments and suggestions.

Thank you very much for your comments and your precious time to review our manuscript. First of all, we have had our manuscript checked by a professional proofreader.

1. Title: please replace “as a ground”. I recommend “ as an important asset”, “as a resource...”, “ as an important foundation”, etc.

We agree with this suggestion and have changed the title into ‘*Cell culture-based shark karyotyping as a ~~ground~~-resource for chromosome-scale genome analysis*’.

2. One of the important achievements of this study was optimizing cell culture condition for sharks. This is not sufficiently detailed and highlighted in Results. Please elaborate.

Thank you very much for your constructive suggestion. We agree with that and have included our basic technical policy in a new paragraph placed at the beginning of the section **Cell culture from shark tissues in Results** as below. Also, we have moved the record of past attempts of shark cell culture initially included in **Discussion** to **Introduction** after some elaboration. Besides, we have tabulated medium formulation in our protocol and previous studies on shark cells (**Supplementary Table 2**). Moreover, to enhance the clarity and self-contained utility of our protocol, we have inserted all names of the different growth factors and mitogens in **Supplementary Fig. 1**. We understand that this reorganization will guide better the readers to prior knowledge of the basis of our protocol derivation.

‘The long-term infeasibility of high-fidelity shark cell culture was thought to be attributable to insufficient adaptation of the culture medium to body fluid osmolality, as mentioned above. To overcome this problem, we referred to past attempts and tried novel combinations of medium ingredients. For fibroblast culture, we derived culture medium supplemented with urea, NaCl and three kinds of cell growth factors (insulin-transferrin-selenium [ITS-G], epidermal growth factor and fibroblast growth factor) at the concentrations used in previous studies^{22,28,30} (Supplementary Fig. 1, Supplementary Table 2). For lymphocyte culture, we adopted culture medium supplemented with the combination of four mitogens, Con A, LPS, PHA and PMA, at the concentrations used in previous studies^{23,25,34} (Supplementary Fig. 1, Supplementary Table 2).’

To systematically examine multiple conditions of cell culture, we need sufficient quantity of biological materials sampled from the same individual under a uniform condition. Unlike teleost fishes, many chondrichthyan species are not egg-laying, and even egg-laying species exhibit low frequency and number of oviposition, allowing little possibility for sampling sufficient embryonic tissues at one time. Availability of live juvenile and adult chondrichthyans is also much smaller. Our first-ever karyotyping of the whale shark was made possible by collaboration with public aquariums, but sampling opportunity of fresh blood at them was highly limited. For these reasons, we could not perform systematic optimization of cell culture protocol with varying conditions under uniform experiment setting. The difficulty of experiment outlined above has exactly been the reason why chromosome studies on chondrichthyans are way behind. In this study, even without thorough optimization of experimental conditions, we successfully introduce our protocol by integrating promising conditions from past studies by other groups. The karyotyping results presented for two more species in the newly included **Supplementary Fig. 5**, as well as the originally included figures, prove

the wide utility of our protocol. Besides, we reworded some parts of the manuscript to express our effort with ‘derivation’ of a new protocol rather than ‘optimization’.

3. The symbol for cytochrome oxidase subunit 1 should be COX1 not COI.

We have modified the text as suggested. Thank you very much for pointing this out.

4. Page 5, “CGH patterns...”: CGH must be explained at its first appearance and it is “Comparative Genomic Hybridization” not “...Genome...”.

We have modified the text as suggested.

5. Page 7, line 3: should be “...relatively high diploid chromosome number..”, not “large”

We have modified the text as suggested.

6. Page 7, line 17: should be “hermaphroditism” not “hermaphrodism”

I appreciate your correction. We have modified the text as suggested.

7. Page 7, line 33: should be “differentiated X and Y sex chromosomes”.

We have modified the text as suggested.

8. Page 8, lines 9-11: the sentence “A karyotype instructs the species’ inherent chromosome number and sizes, as well as centromere positions and provides an ultimate goal of whole-genome sequencing.” reads odd. Please reword.

We have changed this sentence as below:

‘~~A~~The karyotype ~~instructs the~~ provides information about a species’ inherent chromosome number and sizes, as well as centromere positions, and provides an ultimate goal of whole-genome ~~sequencing~~ sequence reconstruction.’

9. Fig. 1 needs a note whether these photos are taken by the authors or retrieved from internet. The latter probably requires permission or at least a webaddress.

The photos in **Fig. 1** were all taken by us the authors (Shigehiro Kuraku, Itsuki Kiyatake, or Nobuyuki Higashiguchi). We have not used these photos in any other publications and media.

10. Supp Fig. 1: should be “supernatant” not “supermatant”.

We have modified the text as suggested.

Response to Reviewer #3:

There are many typographical and grammatical errors in the text.

Thank you very much for your comments and your precious time to review our manuscript. We have had our revised manuscript checked by a professional proofreader.

The authors refer to epidermal cells but probably intended to say epithelial cells. We have modified the text as suggested.

We have modified the text as suggested.

While the karyograms are somewhat informative, the authors ought to show the corresponding giemsa metaphase spreads as well to show the overall quality of the spreads.

Thank you for pointing this out. We have inserted the Giemsa-stained chromosome metaphase spreads (**Supplementary Fig. 3**) from which male karyotypes in **Fig. 3** and female karyotypes in **Supplementary Fig. 2** are derived.

As effective as the modified culture media is for generating dividing cells for chromosome preparation, it is cost-prohibitive for many labs who may want to routinely karyotype chondrichthyan species.

To respond to this suggestions and the comments from **Reviewer #2**, the text has been substantially revised with more detailed information of our policy of protocol derivation in the section **Cell culture from shark tissues** of **Results**. We have also

included more description of past trials of shark cell culture by other researchers in **Introduction** which served as a baseline of identifying the promising experiment conditions.

In this study for fibroblast and lymphocyte culture, we adopted the novel combination of the cell growth factors and mitogens which were used in the previous reports (ref. 20: Garner 1988, ref. 21: Grogan & Lund 1990, ref. 22: Hartmann et al. 1992, ref. 23: McKinney 1992, ref. 24: Poyer & Hartmann 1992, ref. 25: Maddock & Schwartz 1996, ref. 26: Walsh & Luer 1998, ref. 27: Walsh et al. 2006, ref. 28: Parton et al. 2007, ref. 29: Barnes et al. 2008, ref. 30: Uno et al. 2008, ref. 34: Fujiwara et al. 2001). These reagents are widely used for mammalian cell culture and can be easily obtained. To allow direct comparisons of culture media, we have tabulated the combinations of supplements employed in our present and past studies (**Supplementary Table 2**).

As another candidate of supplements, shark yolk extract was used for shark cell culture by Parton et al. (2007). But, we understand that shark yolk extract is available only in some regions of the world and is sometimes prepared from eggs of shark species that have low fecundity and are thus endangered and protected. For these economical and ecological reasons, we are proposing the fibroblast culture media that contain no shark yolk extract. Collectively, we understand that our culture media and conditions allow many researchers to perform cartilaginous fish cell culture with low barriers. So, we have added the following texts to the second paragraph of **Discussion**.

‘Moreover, in that study, the culture medium was supplemented with shark yolk extract, which, to our knowledge, is not widely available²⁸. In addition, our protocol does not require expensive or inaccessible materials, such as shark yolk extract.’

Has the group empirically tested different conditions and formulations? How did they come up with this formulation to begin with.

To systematically test multiple conditions of cell culture, we need sufficient quantity of biological materials sampled from the same individual under a uniform condition. Unlike teleost fishes, many chondrichthyan species are not egg-laying, and even egg-laying species exhibit low frequency and number of oviposition, allowing little possibility for sampling sufficient embryonic tissues at one time. Availability of live juvenile and adult chondrichthyans is also much smaller. Our first-ever karyotyping of the whale shark was made possible by collaboration with public aquariums, but sampling opportunity of fresh blood at them was highly limited. For these reasons, we could not perform systematic optimization of cell culture protocol with varying conditions under uniform experiment setting. The difficulty of experiment outlined above has exactly been the reason why chromosome studies on chondrichthyans are way behind. In this study, even

without thorough optimization of experimental conditions, we successfully introduce our protocol by integrating promising conditions from past studies by other groups. The karyotyping results presented for two more species in the newly included **Supplementary Fig. 5**, as well as the originally included figures, prove the wide utility of our protocol. Besides, we reworded some parts of the manuscript to express our effort with ‘derivation’ of a new protocol rather than ‘development’ or ‘optimization’.

REVIEWERS' COMMENTS:

Reviewer #1 (Remarks to the Author):

After reviewing the response to the comments on initial review, I am satisfied that the authors have addressed my main concern regarding the organization of the manuscript and previous disjointed goal of the manuscript. The authors have appropriately removed or re-framed the discussion of sex determination in sharks to fit with the scope of their results and analyses. Now it is clearly a study that identified methodology for cell culture in elasmobranchs that worked in multiple tissue types. The main goal of bringing this methodology to bear was to produce a quality number of cells that could be utilized for accurate karyotyping which is necessary for full validation of draft genome sequencing projects. The authors were able to accomplish this and conduct some secondary analyses with imaging of the karyotypes to identify telomeric and rRNA sequences through FISH. There are still some small editorial comments though the clarity of the text is much improved.

Additional small editorial comments are below:

Line 44, 'produce' should be 'produces'

Sentence ending on line 48 should have a citation, presumably the current citation 19 used regarding shark body fluid osmolality

Line 112 'the' should be changed to 'a'

Lines 116-119 should be moved from results to the discussion

Line 223 there should be a space between egg and cases

Legend for figure 1 should now include a statement that branch lengths do not reflect phylogenetic distance.

Reviewer #2 (Remarks to the Author):

The authors have adequately responded to the critique and made appropriate changes in the manuscript. I do not have any additional comments

Reviewer #3 (Remarks to the Author):

Minor points:

- The title might be misconstrued. This paper primarily describes improvements in cell culture for shark cells and does not really delve into chromosome-scale genome analysis. In general, the focus/scope of the paper could be made much tighter; my opinion is that it should focus on the excellent cytogenetics that the authors have accomplished and how they have improved upon cell culturing conditions. The paper comes across a bit unfocused and lacks a general theme.
- Line 19 This sentence is awkward. Doesn't karyotyping per se imply cytogenetic techniques? Seems redundant.
- Line 21 I think this sentence is also awkward. The lack of cytogenetic studies in chondrichthyans is due to difficulties in proficient cell culture, not to unique osmoregulatory mechanism, though it is a factor in culturing chondrichthyan cells.

- Line 24 Is it method or culture conditions or both?
- Line 29 Is "solution" the best term here?
- Line 34 Should also include how improvements in genetic mapping have also contributed to long range assemblies, "chromosome" length.
- Line 41 Elephant fish should be elephant shark
- Lines 57-59 This sentence starting with "However" is confusing. I think the intent is to state that sequencing information would be more informative if superimposed on high quality karyotype information
- Line 66 "The abundance of chromosomes has been a hurdle in chondrichthyan cytogenetics, but the most crucial obstacle lies in the supply of cultured cells." Somehow this sentence doesn't convey what the authors probably intend to say: that robust cell culture can alleviate the problem with mitotic index and can increase the overall quality of metaphase chromosomes. Similarly, there are parts of the next paragraph that may not be needed.
- Line 100 I suggest splitting into two sentences, the second one emphasizing the "potential utility" of the method.

Response to Reviewer #1:

Line 44, 'produce' should be 'produces'

We have modified the text as suggested.

Sentence ending on line 48 should have a citation, presumably the current citation 19 used regarding shark body fluid osmolality

As suggested, we have cited the reference (ref.8: Griffith 1981) at the end of this sentence.

Line 112 'the' should be changed to 'a'

We have modified the text as suggested.

Lines 116-119 should be moved from results to the discussion

We have moved this and the next sentences to the first paragraph of **Discussion** as suggested.

Line 223 there should be a space between egg and cases

We have modified the text as suggested.

Legend for figure 1 should now include a statement that branch lengths do not reflect phylogenetic distance.

We have added the sentence below in Figure legend of **Fig. 1** as suggested.

'The branch lengths do not reflect phylogenetic distance.'

Response to Reviewer #3:

The title might be misconstrued. This paper primarily describes improvements in cell culture for shark cells and does not really delve into chromosome-scale genome analysis. In general, the focus/scope of the paper could be made much tighter; my opinion is that it should focus on the excellent cytogenetics that the authors have accomplished and how they have improved upon cell culturing conditions. The paper comes across a bit unfocused and lacks a general theme.

Thank you for the suggestion. The editor suggested the title ‘*Cell culture-based karyotyping of orectolobiform sharks for chromosome-scale genome analysis*’, and we have included this title in the revised manuscript. To reflect your suggestion, we would propose another title ‘*Cell culture-based karyotyping of orectolobiform sharks*’ to the editor and wait for the editor’s response.

Line 19 This sentence is awkward. Doesn’t karyotyping per se imply cytogenetic techniques? Seems redundant.

We have modified the sentence below.

‘Karyotypes, traditionally characterized with cytogenetic techniques, are indispensable for validating genome assemblies whose sequence lengths scale up to chromosome sizes with modern methods—and is traditionally performed using cytogenetic techniques.’

Line 21 I think this sentence is also awkward. The lack of cytogenetic studies in chondrichthyans is due to difficulties in proficient cell culture, not to unique osmoregulatory mechanism, though it is a factor in culturing chondrichthyan cells.

Thank you very much for your suggestion. We have deleted “the unique osmoregulatory mechanism” in this sentence below.

‘Karyotype reports of chondrichthyans are scarce, ~~mainly~~ because of the difficulty in cell culture ~~their unique osmoregulatory mechanism, which hinders cell culture.~~’

Line 24 Is it method or culture conditions or both?

This sentence did not convey our intention. We have modified this sentence below.

‘The utility of the cultured cells enabled the ~~Using this method and culture condition, we performed~~ high-fidelity characterization of their karyotypes, ...’

Line 29 Is “solution” the best term here?

We have changed ‘*technical solution*’ to ‘*protocols*’.

Line 34 Should also include how improvements in genetic mapping have also contributed to long range assemblies, “chromosome” length.

Thank you for your suggestion. We have inserted the sentence about genetic linkage mapping and cited a new reference (ref. 4: Kasahara et al. 2007).

‘Previously, this was achieved only by genetic linkage mapping, performed with a number of offspring (ref. 4: Kasahara et al. 2007). In controlling resultant chromosome-scale sequences, karyotyping serves the final goal by providing information about the number of chromosomes. Karyotype information serves as the final goal by providing information about the number of chromosomes.’

New citations:

4. Kasahara, M. et al. The medaka draft genome and insights into vertebrate genome evolution. *Nature* **447**, 714–719 (2007).

Line 41 Elephant fish should be elephant shark

We are quite conscious about this point. The researchers in Australia, the habitat of this fish, have used the common name ‘elephant fish’. In the revised manuscript, we have maintained ‘elephant fish’ and supplied ‘elephant shark’ as an alias.

Lines 57-59 This sentence starting with “However” is confusing. I think the intent is to state that sequencing information would be more informative if superimposed on high quality karyotype information

That is what we meant. Sequence information is more reliable if confirmed with karyotype information.

• Line 66 “The abundance of chromosomes has been a hurdle in chondrichthyan cytogenetics, but the most crucial obstacle lies in the supply of cultured cells.” Somehow this sentence doesn’t convey what the authors probably intend to say: that robust cell culture can alleviate the problem with mitotic index and can increase the overall quality of metaphase chromosomes. Similarly, there are parts of the next paragraph that may not be needed.

Thank you for the suggestion. We have modified the sentence below.

‘The abundance of chromosomes has been a hurdle in chondrichthyan cytogenetics, but the most crucial obstacle lies in the supply of cultured cells to obtain high-quality chromosome spreads.

Some parts in the next paragraph may not be necessary to experts, but we understand that they can give useful information to those who are not familiar with cytogenetics, such as genome informaticians.

- Line 100 I suggest splitting into two sentences, the second one emphasizing the “potential utility” of the method.

We have modified the sentence below.

‘Using the cultured cells, we have revealed the karyotypes and ~~demonstrated~~ performed FISH mapping for these species-as well-as. Moreover, we have demonstrated the potential utility of our method for modern genomic studies.’